# Correlation between Radiological Characteristics, PET-CT and Histological Subtypes of Primary Lung Adenocarcinoma—A 102 Case Series Analysis

**DOI:** 10.3390/medicina60040617

**Published:** 2024-04-10

**Authors:** Nikola Colic, Ruza Stevic, Mihailo Stjepanovic, Milan Savić, Jelena Jankovic, Slobodan Belic, Jelena Petrovic, Nikola Bogosavljevic, Dejan Aleksandric, Katarina Lukic, Marko Kostić, Dusan Saponjski, Jelena Vasic Madzarevic, Stefan Stojkovic, Maja Ercegovac, Zeljko Garabinovic

**Affiliations:** 1Center for Radiology and MR, University Clinical Center of Serbia, 11000 Belgrade, Serbia; 2Medical Faculty, University of Belgrade, 11000 Belgrade, Serbiajjelena1984@gmail.com (J.J.); majaerce@verat.net (M.E.); 3Clinic for Pulmonology, University Clinical Center of Serbia, 11000 Belgrade, Serbia; 4Clinic for Thoracic Surgery, University Clinical Center of Serbia, 11000 Belgrade, Serbia; 5Center for Nuclear Medicine with PET, University Clinical Center of Serbia, 11000 Belgrade, Serbia; 6Institute for Orthopedics “Banjica”, 11000 Belgrade, Serbia; 7Clinic for Gastroenterohepatology, University Clinical Center of Serbia, 11000 Belgrade, Serbia

**Keywords:** computed tomography, positron emission tomography, diagnostics, minimally invasive adenocarcinoma

## Abstract

*Background and Objectives:* Lung cancer is the second most common form of cancer in the world for both men and women as well as the most common cause of cancer-related deaths worldwide. The aim of this study is to summarize the radiological characteristics between primary lung adenocarcinoma subtypes and to correlate them with FDG uptake on PET-CT. *Materials and Methods:* This retrospective study included 102 patients with pathohistologically confirmed lung adenocarcinoma. A PET-CT examination was performed on some of the patients and the values of SUVmax were also correlated with the histological and morphological characteristics of the masses in the lungs. *Results:* The results of this analysis showed that the mean size of AIS-MIA (adenocarcinoma in situ and minimally invasive adenocarcinoma) cancer was significantly lower than for all other cancer types, while the mean size of the acinar cancer was smaller than in the solid type of cancer. Metastases were significantly more frequent in solid adenocarcinoma than in acinar, lepidic, and AIS-MIA cancer subtypes. The maximum standardized FDG uptake was significantly lower in AIS-MIA than in all other cancer types and in the acinar predominant subtype compared to solid cancer. Papillary predominant adenocarcinoma had higher odds of developing contralateral lymph node involvement compared to other types. Solid adenocarcinoma was associated with higher odds of having metastases and with higher SUVmax. AIS-MIA was associated with lower odds of one unit increase in tumor size and ipsilateral lymph node involvement. *Conclusions:* The correlation between histopathological and radiological findings is crucial for accurate diagnosis and staging. By integrating both sets of data, clinicians can enhance diagnostic accuracy and determine the optimal treatment plan.

## 1. Introduction

Lung cancer is the second most common form of cancer in the world for both men and women as well as the most common cause of cancer-related deaths worldwide [1]. Around 2 million new cases of lung cancer are discovered in the world every year, with an increasing trend each year. In total, 85% of all diagnosed cases of lung cancer are non-small pathohistological types, and about 45% of them are lung adenocarcinomas [2].

The International Association for the Study of Lung Cancer (IASLC), the American Thoracic Society (ATS), and the European Respiratory Society (ERS) published in 2011 a multidisciplinary classification of lung adenocarcinomas (ADCs), resulting from a consensus between chest physicians, oncologists, thoracic surgeons, pathologists, molecular biologists, and radiologists [3]. Further refinements were made in the WHO (World Health Organisation) classification from 2015 and 2021, integrating genetic and molecular data.

The radiological presentation of peripheral ADCs has a various spectrum of presentations from subsolid to solid nodules and masses. This wide range of imaging findings turns out to have a good correlation with adenocarcinoma subtypes, histological patterns, as well as prognosis [4,5,6]. Recent advances in imaging techniques, such as positron emission tomography–computed tomography (PET-CT) and multidetector computed tomography (MDCT), have improved the diagnosis, staging, and management of lung adenocarcinoma [7]. This essay aims to explore the histological subtypes of lung adenocarcinoma and their correlation with PET-CT and MDCT findings.

Terminology of lung adenocarcinoma has been significantly revised in the new WHO classification by discontinuing the terms bronchioloalveolar carcinoma (BAC) and mixed subtype adenocarcinoma and adding terms of adenocarcinoma in situ (AIS) as a preinvasive lesion as well as minimally invasive adenocarcinoma (MIA). The subtypes of clear cell and signet ring adenocarcinoma and term mucinous cystadenocarcinoma have been discontinued and later included under the category of colloid adenocarcinoma while keeping five general histological types: acinar, papillary, micropapillary, lepidic, and solid [6,7].

The aim of this study is to evaluate the correlation between morphological characteristics of primary lung adenocarcinoma and histopathological subtypes of lung adenocarcinoma. Furthermore, we aimed to analyze the correlation between the values of the maximum uptake of FDG on the performed PET-CT with certain radiological characteristics and histological subtypes of the tumor.

## 2. Materials and Methods

### 2.1. Patients

This retrospective study included 102 patients with lung adenocarcinoma confirmed via pathohistological examination, starting from 1 January to 31 December 2017 inside the Clinic for thoracic surgery and the Clinic for pulmonology at the University Clinical Center of Serbia. The sample of tumor tissue was obtained for pathohistological evaluation.

### 2.2. CT and FDG PET CT Image Acquisition

All CT examinations were performed with CT scanners after intravenous contrast application in the late arterial phase in all patients. The chest CT features are reviewed by radiologists and include tumor consistency, the size of the tumor (largest diameter in the axial plane in the lung window), shape and margins, as well as the relationship to the surrounding structures (pleura, vascular components, and bronchi). The enlargement of lymph nodes (more than 15 mm in shorter axis) and their localization were monitored – paratracheal, hilar on the same side as the tumor, contralateral, and in the supraclavicular pits. Three patients with predominant micropapillary subtypes of adenocarcinoma reported by pathologists in this group were excluded from the study due to the fact that a small number would not have statistical importance nor would show true radiological characteristics of that tumor subtype; also, neither one of those patients was examined on PET-CT scan.

#### Acquisition and Interpretation of ^18^F-FDG PET/CT Findings

Fluorine-18-FDG PET/CT examination was performed in all patients on a 64-slice hybrid PET/CT scanner (Biograph, TruePoint64, Siemens Medical Solutions, Inc., Malvern, PA, USA) at the National PET Center, University Clinical Center of Serbia. A total of 5.5 MBq/kg of ^18^F-FDG was applied intravenously followed by an hour of resting. Whole body low-dose non-enhanced CT (120 kV, 5 mm slice thickness, pitch 1.5, and rotation time 0.5 s) and PET scans (3 min per field, 6 fields of view) were then performed. Corrected and uncorrected attenuation low dose CT, PET, and fused PET/CT scans were presented on a Syngo Multimodality workstation for interpretation. After excluding benign tumors and regions of a physiological uptake, ^18^F-FDG PET/CT findings were considered in cases of greater accumulation of the tracer within an observed lesion than the accumulation in the great mediastinal blood vessels, surrounding tissue, and liver, which were further analyzed visually and semi-quantitatively. The level of glucose metabolism within the lesion was assessed on reconstructed images using the maximum standardized uptake value (SUVmax), which was calculated by the tracer’s uptake in the region of interest divided by an administrated radioactivity and the patient’s weight.

### 2.3. Statistical Methods

The normality of the distribution of continuous variables was evaluated by using visual inspection of histograms and probability plots. Data were presented as mean ± SD or median (interquartile range [IQR]) for continuous variables, depending on the normality of data distribution, and number (percentage) for categorical variables. Differences in patient and cancer characteristics between the five groups of cancer types were assessed using the ANOVA or Kruskal–Wallis test for continuous data and the Chi-square test for categorical data. To adjust for multiple comparisons, Bonferroni correction was applied for all post hoc comparisons. Separate logistic regression analyses were performed to estimate the relationship between patient and cancer characteristics and the occurrence of different cancer types adjusted for age, sex, and smoking status. The “One vs. all” method was used to assess the association between patient and cancer characteristics and certain types of cancer with respect to other cancer types. Odds ratios (OR) with 95%CI were calculated and the Hosmer–Lemeshow goodness-of-fit test was performed to assess the overall model fit [8]. All statistical tests were two-sided and were performed at a 5% significance level or by using a 95% confidence interval generated by the bootstrap method set to 1.000 reiterations. The statistical analysis was performed using SPSS version 23.0 software (SPSS Inc., Chicago, IL, USA). The statistical significance was set at *p* < 0.05.

## 3. Results

Baseline patients and cancer characteristics are summarised in Table 1, Table 2, Table 3, Table 4 and Table 5 and compared between five categories of lung adenocarcinoma.

Table 1 shows a comparison between subtypes of lung adenocarcinoma and gender, age, and smoking status. The results show that patients are predominantly male. It has been determined that the lepidic type of cancer was significantly more common in females than in males (78.9% vs. 21.1%, *p* = 0.003), while the acinar type was significantly more common in males than in females (65.6% vs. 34.4%, *p* = 0.003). There was no significant correlation between smoking status or age group and any type of cancer.

While obtaining pathohistological materials, 24 patients (23.6%) were in stage IA and IB, 34 patients (33.4%) were in stage II (IIA and IIB), and 26 patients (25.4%) were in stage III of the disease (III A,B and stage); 18 patients (17.6%) were in stage IV (IVA and IVB) of the disease.

Table 2 shows a comparison between subtypes of adenocarcinoma and tumor size, components, and CT characteristics of tumor edges. The results indicate that the mean size of AIS-MIA cancer was significantly lower than for all other cancer types, while the mean size of the acinar cancer was smaller than in the solid type of cancer (37.2 ± 7.6 vs. 47.7 ± 12.6, *p* = 0.002). Even though there was no significant statistical correlation, it has been noticed that in the AIS-MIA subtype, ground-glass component is most dominant.

Comparing the involvement of surrounding structures and lymph node metastasis in Table 3, the results show that only in the AIS-MIA subtype there was almost no infiltration of the surrounding structures or lymph node involvement. All other subtypes have similar infiltration of pleura and great vessels.

Characteristics of primary lung adenocarcinoma according to subtypes in relation to the presence of metastases and PET findings, as well as other tumor characteristics are shown in Table 4 and Table 5. Metastases were significantly more frequent in solid adenocarcinoma (61%) than in acinar (9.4%, *p* = 0.001), lepidic (0%, *p* < 0.001), and AIS-MIA (0%, *p* = 0.003) cancer subtypes. The maximum standardized uptake (SUVmax) was significantly lower in AIS-MIA than in all other cancer types, and in acinar compared to solid cancer (4.9 ± 1.1 vs. 6.3 ± 0.8, *p* = 0.001). Papillary adenocarcinoma had higher odds of developing contralateral lymph node involvement compared to other types of cancer (OR 4.49, 95%CI 1.02–19.73). Solid adenocarcinoma was associated with higher odds of having metastases (OR 14.09, 95%CI 3.51–56.41) and with higher SUVmax (OR for one unit increase 2.64, 95%CI 1.48–4.69). AIS-MIA was associated with lower odds of one unit increase in tumor size (OR 0.65 95%CI 0.51–0.83), ipsilateral lymph node involvement (0.20 95%CI 0.05–0.85), and one unit increase in SUVmax (OR 0.07 95%CI 0.02–0.29) with higher odds of ground-glass presentation (OR 7.19, 95%CI 1.35–38.34). There were no significant associations between the selected characteristics and acinar and solid cancer compared to other cancer types.

Some of the CT and PET/CT findings of the cases from the study are shown in the Figure 1, Figure 2, Figure 3, Figure 4 and Figure 5 below.

## 4. Discussion

Keeping in mind the fact that lung cancer is currently one of the most common forms of cancer in the world, and lung adenocarcinoma is the most common histological type of lung cancer, we believe that timely diagnosis significantly improves the outcome of the course of the disease [8,9].

In the literature, similar studies that also correlate radiological features with lung ADC can be found. We have come to conclusions similar to those of Huang et al. regarding the prediction of histological subtypes of lung tumors using radiological features. The conclusions of both studies are that larger studies need to be made in order to optimize and validate these results [10].

Invasive adenocarcinoma is most often seen as a solid nodule, but it may also be partially solid, and occasionally a ground-glass nodule. A lobar pattern of ground-glass opacity (GGO) can be seen in some of the cases. Lobulated tumors in the Ia stage of lung adenocarcinoma correlate with well-differentiated, slowly growing tumors. Thick (≥2 mm) spiculation has been associated with vascular invasion, mediastinal lymphadenopathy, and decreased survival rate. If in the Ia stage lung adenocarcinoma is seen as a partially solid nodule, then an extensive ground-glass component suggests a favorable outcome [10,11,12]. Histologically, the solid component typically corresponds to invasive patterns, while a lepidic pattern is usually seen as the ground-glass component. The absence of pleural retraction in lung adenocarcinoma is also a sign of a favorable prognosis [11]. In solid adenocarcinomas, the presence of nodules, or lobulated edges on thin-section CT has usually been associated with poor differentiation on pathohistological examination and these cases have a much higher risk of an adverse outcome [5,9,11]. A number of studies conducted to this date deal with the significance of determination of the type of lung adenocarcinoma, as well as with its further prognosis based on staging and certain gene mutations. It has been proven that the lepidic type associated with a better outcome in patients with lung adenocarcinoma is also a predictor of survival in numerous papers and morphological characteristics that we have correlated here. Also, the correlation of tumor morphological features with PET-CT results gives a clear picture of the prognosis of the outcome. Although the weakness of this study is that other histological types of non-small cell lung cancer are not included, which gives similar CT characteristics, it still gives its importance, especially in a certain part of patients where surgical treatment is not possible. Then the histopathological type of tumor is obtained by bronchoscopy, tru-cut, or FNA biopsy and the sample is significantly smaller in volume and the correlation with the morphological characteristics on CT is of great importance for making a final diagnosis and prognosis of outcome. Tsutani et al. [12] reported that candidates for adjuvant chemotherapy in Stage I lung adenocarcinoma need to be selected based on the pathological invasive component size. Adjuvant chemotherapy would not be beneficial for patients with AIS or MIA and those with an invasive component size of 5 to 20 mm.

Most findings on larger studies as well as in our group of patients go for CT and PET-CT characteristics of histological subtypes [13,14,15,16,17,18,19]:

Lepidic patterns are often present as ground-glass opacity (GGO) on CT imaging. GGOs typically demonstrate a hazy or cloudy appearance and are associated with a favorable prognosis. This type often exhibits low metabolic activity on PET imaging. This pattern typically manifests as a focal area of increased radiotracer uptake on CT, reflecting the underlying ground-glass opacity or consolidation.

Acinar patterns are associated with a higher likelihood of lymph node involvement and poorer prognosis. The acinar subtype, composed of glandular structures, generally demonstrates moderate to high metabolic activity on PET-CT imaging. PET scans reveal focal areas of increased radiotracer uptake corresponding to solid components within the tumor.

Papillary patterns may manifest as a solid nodule with lobulated margins on CT imaging. The papillary subtype, characterized by papillary projections, typically shows increased radiotracer uptake on PET scans. The presence of avid radiotracer uptake corresponds to the solid components or invasive portions of the tumor, highlighting a higher risk of lymph node metastasis and potential aggressiveness.

Solid patterns typically appear as a homogeneous solid nodule on CT imaging. It is associated with a higher risk of lymph node metastasis, distant spread, and unfavorable prognosis. The solid subtype, composed of sheets of tumor cells without distinctive glandular or papillary structures, generally exhibits high metabolic activity on PET imaging.

And for PET, given the increase in the number of patients from year to year, as well as the beginning of screening for this disease in some countries for risk groups, it is necessary to work in this area as much as possible to see a better radiological picture and contribute to the correlation between different morphological features on CT with pathological, immune, and genetic characteristics, as well as the characteristics that the tumor shows on other imaging methods, all with the aim of better understanding this disease. An example can be the proof that in the first stage of the disease, as well as in patients with AIS and MIA, the five-year survival is almost 100% [6,14,19,20,21,22,23].

There were no other major studies that show gender correlation, and based on our experience, there is no predilection for any gender to develop any subtype of adenocarcinoma, so basic gender results in this paper are the consequence of a relatively small sample [24,25,26].

Elevated standard uptake values (SUVs) on fluorodeoxyglucose positron emission tomography (PET) correlate with cellular proliferation and the aggressiveness of the primary cancer. The sensitivity of PET for AIS is usually very low. PET is commonly used for staging and follow-up of invasive adenocarcinoma, and for lesions of 7 mm or larger, SUV for adenocarcinoma of the lung tends to be lower than for other histologic types of lung cancer and correlate inversely with survival [27,28,29,30,31,32,33]. Larger studies show similar results for smaller lung nodules on PET-CT with no major benefit except for the staging of the disease. The importance of radiological features in detecting invasives of the lesion is crucial in these kinds of cases. New studies also focus on the correlation between epidermal growth factor (EGFR) mutation and standard uptake values (SUV) [26,33].

Our study has limitations as only a small number of patients were included, so the sample size might not be reliable for results, but in comparison to other studies, the results are very similar. The validation of our findings with a larger group is required.

## 5. Conclusions

Radiology has a significant role in the diagnosis and monitoring of the course of the disease, as well as in determining its prognosis, and thus the greatest influence on the clinical decision on the method of treatment. A good example of this may be smaller ground-glass lesions that have been shown to be minimally invasive, and a shorter follow-up may have been advised to rule out another etiology, rather than primary resection. The morphological characteristics of the tumor may indicate to some extent histological types of lung adenocarcinoma, but in correlation with PET-CT, they significantly help to differentiate them when the tumor tissue sample itself is small. CT is also the primary method for monitoring responses to chemotherapy and radiotherapy, as well as for diagnosing disease metastases.

The correlation between histopathological and radiological findings is crucial for accurate diagnosis and staging. By integrating both sets of data, clinicians can enhance diagnostic accuracy and determine the optimal treatment plan. Additionally, the presence of specific histopathological features, such as micropapillary or solid patterns, may indicate a higher risk of lymph node involvement, which can guide the decision for surgical resection or lymph node sampling.

Furthermore, histopathological and radiological correlation is crucial for assessing treatment response and disease progression. Changes in tumor size, density, and metabolic activity observed on follow-up imaging scans can help evaluate the effectiveness of treatment modalities, such as chemotherapy or targeted therapy. If there is discordance between the radiological and histopathological findings, additional investigations, such as repeated biopsies or molecular testing, may be necessary to guide treatment adjustments.

Histopathological and radiological correlation plays a fundamental role in the management of lung adenocarcinoma. The integration of histopathological findings with radiological imaging allows accurate diagnosis, staging, treatment planning, and assessment of treatment response. A multidisciplinary approach involving pathologists, radiologists, and clinicians is essential in optimizing patient care and improving outcomes in individuals with lung adenocarcinoma.

## Figures and Tables

**Figure 1 medicina-60-00617-f001:**
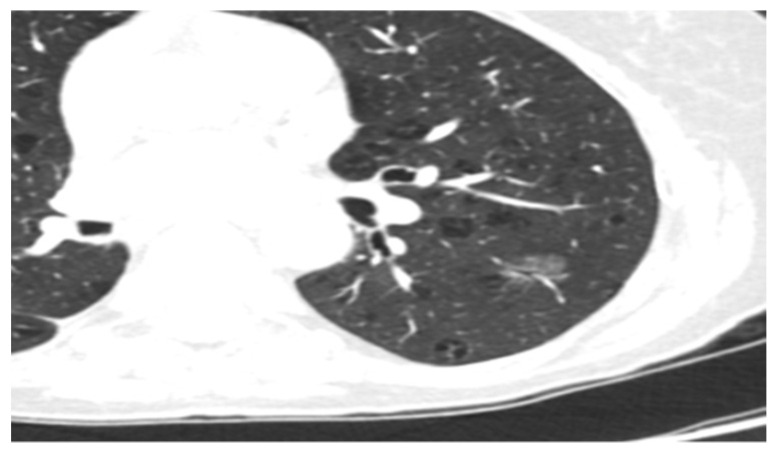
Adenocarcinoma in situ (AIS) ground-glass nodule.

**Figure 2 medicina-60-00617-f002:**
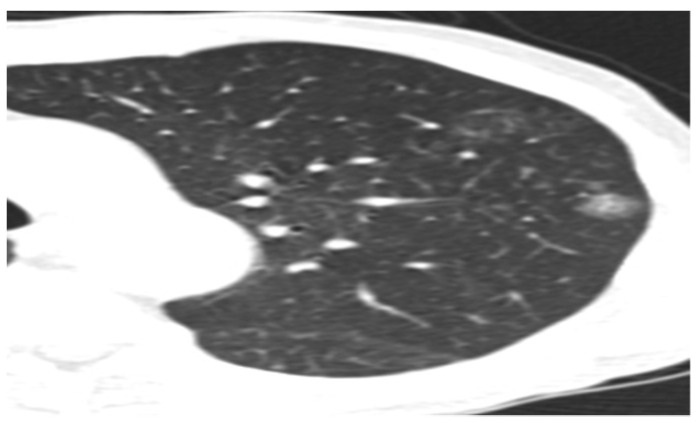
Minimally invasive adenocarcinoma (MIA).

**Figure 3 medicina-60-00617-f003:**
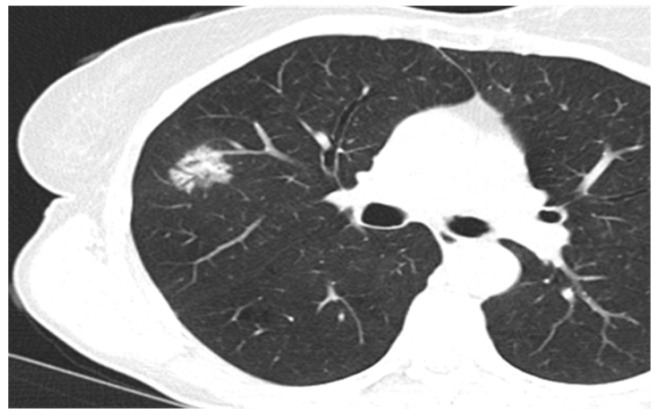
Lepidic type of adenocarcinoma.

**Figure 4 medicina-60-00617-f004:**
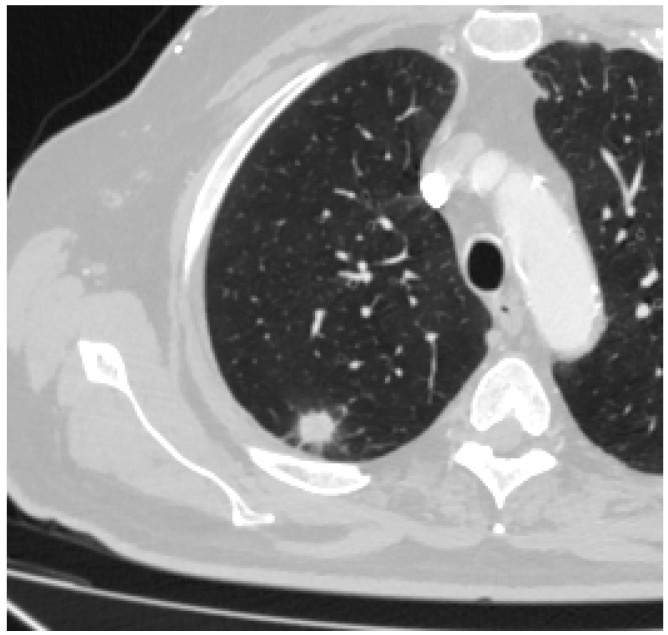
Spiculation of tumor.

**Figure 5 medicina-60-00617-f005:**
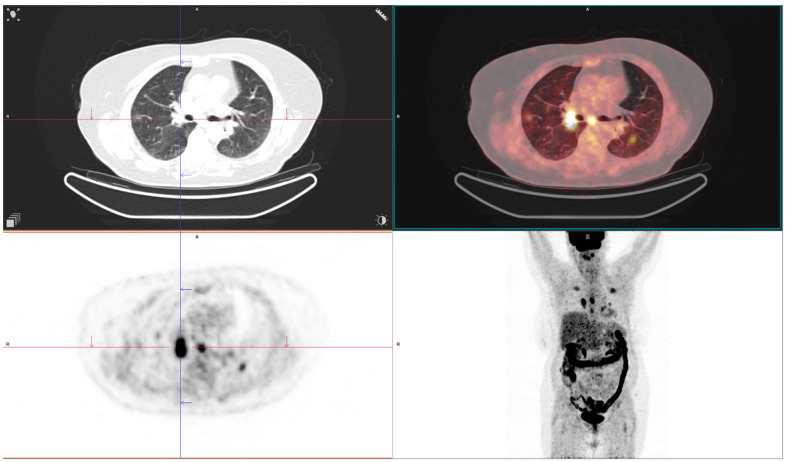
PET CT of solid lung lesion on the **right** and ground-glass lession in the **left** lung with different FDG uptake.

**Table 1 medicina-60-00617-t001:** Characteristics of primary lung adenocarcinoma by subtype in relation to gender, age and smoking status.

	Acinar	Papillary	Lepidic	Solid	AIS-MIA	*p* Value	Comparison Group *	Post Hoc *p* Value ^¥^
	*n* = 32 (31.4%)	*n* = 28 (27.5%)	*n* = 19 (18.6%)	*n* = 13 (12.7%)	*n* = 10 (9.8%)			
Age, mean ± SD	62.8 ± 7.0	62.7 ± 7.0	61.8 ± 7.4	63.7 ± 7.2	61.0 ± 5.6	0.893		
Gender, *n* (%)								
Male	21 (65.6)	14 (50.0)	4 (21.1)	9 (69.2)	5 (50.0)	**0.024**	Acinar vs. Lepidic	0.003
Female	11 (34.4)	14 (50.0)	15 (78.9)	4 (30.8)	5 (50.0)			
*p* value	*p* > 0.005	*p* > 0.005	*p* > 0.005	*p* > 0.005	*p* > 0.005			
Smoking status, *n* (%)								
Non-smoker	10 (31.3)	16 (57.1)	5 (26.3)	7 (53.8)	1 (10.0)	0.052		
Former smoker	8 (25.0)	2 (7.1)	2 (10.5)	1 (7.7)	1 (10.0)			
Current smoker	14 (43.8)	10 (35.7)	12 (63.2)	5 (38.5)	8 (80.0)			
*p* value	*p* > 0.005	*p* > 0.005	*p* > 0.005	*p* > 0.005	*p* > 0.005			

ANOVA test was used for continuous variables and the results are presented in the table above. * bootstrapped at 1000 iterations. ^¥^ Bonferroni correction was applied for multiple comparisons (0.05/10 comparisons = 0.005). Interpretation: overall *p* value indicates whether there is an overall significant difference between these 5 categories. For those comparisons that are significant in the overall comparison (bold, *p* < 0.05), a post hoc analysis was performed to see exactly where the difference lies. Since there are a lot of comparisons because there are 5 categories to compare, the only ones left are those that are significant (*p* < 0.005 and not <0.05 due to multiple comparisons).

**Table 2 medicina-60-00617-t002:** Characteristics of primary lung adenocarcinoma by subtype in relation to tumor size, component, and edges.

	Acinar	Papillary	Lepidic	Solid	AIS-MIA	Overall *p* Value	Comparison Group	Mean Difference
	*n* = 32	*n* = 28	*n* = 19	*n* = 13	*n* = 10			
Tumor size, mean ± SD	37.2 ± 7.6	41.8 ± 8.6	38.2 ± 6.0	47.7 ± 12.6	24.9 ± 3.7	<0.001	Acinar vs. Solid	10.44
							Acinar vs. AIS-MIA	12.35
							Papillary vs. AIS-MIA	16.89
							Lepidic vs. AIS-MIA	13.26
							Solid vs. AIS-MIA	22.79
Component *n* (%)								
Solid	32 (100)	28 (100)	19 (100)	13 (100)	9 (90.0)	0.054		
Necrosis	3 (9.4)	9 (32.1)	5 (26.3)	4 (30.8)	0 (0.0)	0.074		
Ground glass	3 (9.4)	0 (0.0)	1 (5.3)	1 (7.7)	3 (30.0)	0.051		
*p* value	*p* > 0.005	*p* > 0.005	*p* > 0.005	*p* > 0.005	*p* > 0.005	*p* > 0.005		
Edges n (%)								
Round	19 (59.4)	14 (50.0)	14 (73.7)	7 (53.8)	5 (50.0)	0.244		
Lobular	4 (12.5)	4 (14.3)	2 (10.5)	5 (38.5)	3 (30.0)			
Spiculated	9 (28.1)	10 (35.7)	3 (15.8)	1 (7.7)	2 (20.0)			
*p* value	*p* > 0.005	*p* > 0.005	*p* > 0.005	*p* > 0.005	*p* > 0.005			

ANOVA test was used for continuous variables, and the results are presented in the table above.

**Table 3 medicina-60-00617-t003:** Characteristics of primary lung adenocarcinoma according to subtypes in relation to the involvement of surrounding structures and involvement of lymph nodes.

	Acinar	Papillary	Lepidic	Solid	AIS-MIA	Overall *p* Value
	*n* = 32	*n* = 28	*n* = 19	*n* = 13	*n* = 10	
Pleural involvement, *n* (%)	11 (34.4)	15 (53.6)	5 (26.3)	8 (61.5)	2 (20.0)	0.084
Bronchial cut-off, *n* (%)	12 (37.5)	13 (46.4)	10 (52.6)	9 (69.2)	5 (50.0)	0.41
Vascular invasion, *n* (%)	11 (34.4)	16 (57.1)	9 (47.4)	6 (46.2)	3 (30.0)	0.397
No lymph node involvement	9 (28.1)	4 (14.3)	9 (47.7)	2 (15.4)	7 (70.0)	0.049
Ipsilateral lymph node involvement	18 (56.3)	18 (64.3)	8 (42.1)	9 (69.2)	3 (30.3)	
Contralateral lymph node involvement	5 (15.6)	6 (21.4)	2 (10.5)	2 (15.4)	0 (0.0)	
*p* values	*p* > 0.005	*p* > 0.005	*p* > 0.005	*p* > 0.005	*p* > 0.005	

Kurksal–Wallis test was used for continuous variables, and the results are presented in the table above. Bonferroni correction was applied for multiple comparisons (0.05/10 comparisons = 0.005).

**Table 4 medicina-60-00617-t004:** Characteristics of primary lung adenocarcinoma according to subtypes in relation to the presence of metastases and PET findings.

	Acinar	Papillary	Lepidic	Solid	AIS-MIA	Overall *p* Value	Comparison Group	Mean Difference	95%CI **	Post Hoc *p* Value ^¥^
	*n* = 32	*n* = 28	*n* = 19	*n* = 13	*n* = 10					
Metastases present, *n* (%)	3 (9.4%)	7 (25.0%)	0 (0.0%)	8 (61.5%)	0 (0.0%)	**<0.001**	Acinar vs. solid	na	na	0.001
							Lepidic vs. solid	na	na	<0.001
							Solid vs. AIS-MIA	na	na	0.003
SUVmax, mean ± SD	4.9 ± 1.1	5.3 ± 1.3	5.1 ± 0.7	6.3 ± 0.8	3.3 ± 0.8	**<0.001**	Acinar vs. solid	−1.35	−1.89 to −0.76	0.001
							Acinar vs. AIS-MIA	1.65	1.00 to 2.28	<0.001
							Papillary vs. AIS-MIA	2.01	1.35 to 2.72	<0.001
							Lepidic vs. AIS-MIA	1.83	1.23 to 2.38	<0.001
							Solid vs. AIS-MIA	−3	2.32 vs. 3.59	<0.001

Chi-square test was used for numerical variables, and the results are presented in the table above. Adjusted for age, sex, and smoking status. Bolded values are significant. Na, not applicable. ** bootstrapped at 1000 iterations. ^¥^ Bonferroni correction was applied for multiple comparisons (0.05/10 comparisons = 0.005). Interpretation: overall *p* value indicates whether there is an overall significant difference between these 5 categories. For those comparisons that are significant in the overall comparison (bold, *p* < 0.05), a post hoc analysis was performed to see exactly where the difference lies. Since there are a lot of comparisons because there are 5 categories to compare, the only ones left are those that are significant (*p* < 0.005 and not <0.05 due to multiple comparisons).

**Table 5 medicina-60-00617-t005:** Odds ratios with bootstrapped 95%CI of the adjusted relationship of cancer characteristics with type of cancer.

	Acinar	Papillary	Lepidic	Solid	AIS-MIA
	*n* = 32	*n* = 28	*n* = 19	*n* = 13	*n* = 10
Characteristic	OR (95%CI)	OR (95%CI)	OR (95%CI)	OR (95%CI)	OR (95%CI)
Tumor size	0.97 (0.92–1.02)	1.04 (1.00–1.09)	1.00 (0.95–1.05)	**1.11 (1.04–1.18)**	**0.65 (0.51–0.83)**
Necrosis	0.27 (0.07–1.03)	2.57 (0.90–7.37)	1.69 (0.46–6.17)	1.80 (0.47–6.96)	0
Ground glass	1.25 (0.27–5.89)	0	0.69 (0.07–6.59)	1.00 (0.11–9.38)	**7.19 (1.35–38.34)**
Round edges	1.0	1.0	1.0	1.0	1.0
Lobular edges	0.62 (0.18–2.22)	9.91 (0.25–3.22)	0.32 (0.06–1.67)	3.17 (0.83–12.19)	2.28 (0.48–10.81)
Spiculated edges	1.16 (0.42–3.16)	2.16 (0.79–5.89)	0.43 (0.11–1.74)	0.28 (0.03–2.42)	1.00 (0.18–5.62)
Pleural involvement	0.62 (0.25–1.53)	2.18 (0.89–5.34)	0.52 (1.16–1.66)	2.48 (0.73–8.43)	0.35 (0.70–1.77)
Bronchial cut-off	0.60 (0.25–1.48)	0.87 (0.35–2.16)	0.90 (0.31–2.62)	3.53 (0.93–13.36)	1.17 (0.30–4.56)
Vascular invasion	0.55 (2.23–1.33)	2.06 (0.85–4.99)	1.17 (0.41–3.34)	1.11 (0.34–3.60)	0.52 (0.13–2.17)
No lymph node involvement	1.0	1.0	1.0	1.0	1.0
Ipsilateral lymph node involvement	1.08 (0.40–2.90)	3.26 (0.98–10.80)	0.43 (1.14–1.34)	2.54 (0.50–12.98)	**0.20 (0.05–0.85)**
Contralateral lymph node involvement	1.32 (0.34–5.16)	**4.49 (1.02–19.73)**	0.30 (0.05–1.74)	2.34 (0.29–19.04)	0
Metastases present	0.34 (0.09–1.33)	1.93 (0.65–5.72)	0	**14.09 (3.51–56.41)**	0
SUVmax	0.86 (0.59–1.23)	1.21 (0.86–1.73)	1.04 (0.69–1.57)	**2.64 (1.48–4.69)**	**0.07 (0.02–0.29)**

Interpretation: If the OR is less than 1, it means that the characteristic is less present in that cancer than in others, if the OR is greater than 1, it means that this characteristic is more present in that tumor than in the others. If 95% of the CI does not contain 1, it means that the difference is statistically significant (bold in each case).

## Data Availability

The data that support the findings of this study are available from the first author (N.C.) upon reasonable request.

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
