# Peer review of "Correlation between Radiological Characteristics, PET-CT and Histological Subtypes of Primary Lung Adenocarcinoma—A 102 Case Series Analysis"

_medicina, 2024, doi:10.3390/medicina60040617_

Round 1

Reviewer 1 Report

Comments and Suggestions for Authors

This paper provides an insightful examination into how radiological imaging, specifically PET-CT, correlates with the histological subtypes of lung adenocarcinoma. This retrospective analysis included 102 patients, focusing on the relationship between imaging features and pathohistological findings to potentially guide diagnosis and treatment strategies.

The small sample size and its retrospective nature may affect the generalizability of the results, as acknowledged by the authors. They suggest the need for larger, prospective studies to validate these preliminary findings further.

Comments on the Quality of English Language

Minor editing of English language required

Author Response

Dear Reviewer,

Thank You very much for taking time to review our paper and give us feedback.

We made revisions as you stated and highlighted it in text (Bolded text).

English language was edited by native English speaker and corrections were made.

As we stated, this study was on a small group and suggested larger prospective studies should be done to correlate our results.

Thank you very much for your kind and expert revision.

Reviewer 2 Report

Comments and Suggestions for Authors

Dear Authors,

Thanks for your efforts in this field. In fact, the data on PET-CT and GGO is lacking. The histological subtype is detailed. However, some conclusions were not novel. For example, solid adenocarcinoma was associated with higher odds of having metastases and with higher SUVmax. 

The method was clear. 

The conclusion based on the result was solid.

I suggest focusing on the new findings of this study in the discussion section.

Author Response

Dear Reviewer,

Thank You very much for taking time to review our paper and give us feedback.

We made revisions as you stated and highlighted it in text (Bolded text).

In discussion section we added more data to correlate with new studies on this field, such is corelation of PET-CT and EGFR mutations in lung adenocarcinoma, and cost/benefit ratio in doing PET-CT in early stage of lung cancer.

Thank you very much for your kind and expert revision.

Reviewer 3 Report

Comments and Suggestions for Authors

This is a single center retrospective study on the radiological characteristics between primary lung adenocarcinoma subtypes and also correlate them with FDG uptake on PET-CT. The authors reported that the mean size of AIS-MIA cancer was significantly lower than for all other cancer types, while mean size of the acinar cancer was smaller than in solid type of cancer. Metastases were significantly more frequent in solid adenocarcinoma than in acinar, lepidic and AIS-MIA cancer subtypes. The maximum standardized FDG uptake was significantly lower in AIS-MIA than in all other cancer types, and in acinar predominant subtype compared to solid cancer. This study may provide some useful information on the correlation between radiological characteristics, PET-CT and histological subtypes of primary lung adenocarcinoma. I have some comments.

<Comments>

1. Please describe the conclusion section in the abstract.

2. In line 44, Please describe the full term of the abbreviation, “WHO.

3. In the result section, there is no information about the patient's final stage of lung cancer. If you have data, please add the information of final stage.

4. Were there cases where multiple types of lung adenocarcinoma coexist in one patient?

Comments on the Quality of English Language

Minor editing of English language required.

Author Response

Dear Reviewer,

Thank You very much for taking time to review our paper and give us feedback.

We made revisions as you stated and highlighted it in text (Bolded text).

  1. We made change in abstract to describe conclusion as you stated.
  2. We made correction as you stated.
  3. We added in our results stage of patient at time when pathohistological material was obtained.
  4. In some cases there were multiple histological types of lung adenocarcinoma in one tumour we used predominant type that was described in pathohistological report (more than 50% of cells).

English language was edited by native English speaker and corrections were made.

Thank you very much for your kind and expert revision.